# Highly selective transition-metal-free transamidation of amides and amidation of esters at room temperature

Guangchen Li [1] & Michal Szostak [1]

Amide chemistry has an essential role in the synthesis of high value molecules, such as pharmaceuticals, natural products, and fine chemicals. Over the past years, several examples of transamidation reactions have been reported. In general, transition-metal-based catalysts or harsh conditions are employed for these transformations due to unfavorable kinetics and thermodynamics of the process. Herein, we report a significant advance in this area and present the general method for transition-metal-free transamidation of amides and amidation of esters by highly selective acyl cleavage with non-nucleophilic amines at room temperature. In contrast to metal-catalyzed protocols, the method is operationally-simple, environmentally-friendly, and operates under exceedingly mild conditions. The practical value is highlighted by the synthesis of valuable amides in high yields. Considering the key role of amides in various branches of chemical science, we envision that this broadly applicable method will be of great interest in organic synthesis, drug discovery, and biochemistry.

[1] Department of Chemistry, Rutgers University, 73 Warren Street, Newark, NJ 07102, USA. Correspondence and requests for materials should be addressed to M.S. (email: michal.szostak@rutgers.edu)

The amide bond is the most fundamental functional group in chemistry and biology[1–3]. The amide linkage is essential for the structure of peptides, proteins and other biologically-relevant molecules[4–6]. The amide bond is found in numerous natural products, pharmaceuticals and polymers[7]. Recent surveys indicate that the amide bond is present in 25% of registered drugs, while amidation reactions represent the most common reaction performed in the synthesis of pharmaceuticals[8]. As a consequence of the ubiquity of amide bonds, methods for the synthesis of amides have a major impact on every branch of chemical science.

Transamidation reactions are potentially the most useful class of amide bond forming reactions in organic chemistry (Fig. 1)[9,10]. However, while different reagents have been developed for these transformations, the chemoselective transition-metal-free[11–14] transamidation under mild conditions remains an unsolved challenge. Slower to develop have also been methods that allow for the direct conversion of esters to amides[10]. Among many potential advantages, the direct transamidation of amides and esters offers the enhanced stability of precursors, compatibility with multistep reaction sequences, and direct engagement of the typically-considered unreactive amide and ester functional groups in their latent form. For example, in their recent studies, Hillmyer and co-workers employed metal-free transamidation of activated poly(N,N-Boc$_2$-amide) derived directly from primary acrylamide to enable the synthesis of polymers by exploiting postpolymer modification (*vide infra*)[15]. The recent developments notwithstanding, this specific method is incompatible with the extremely valuable non-nucleophilic amines.

Meanwhile, there have been major developments in the discovery of transition-metal-catalyzed cross-coupling reactions of amides and aromatic esters by N–C(acyl) and O–C(acyl) cleavage. This powerful reaction manifold[16] demonstrates that common amides and esters can be readily modulated to provide carbon–carbon and carbon–heteroatom bond functionalization reactions[17–21] for chemical synthesis through resonance destabilization of the amide and ester bond[22,23]. The broad success of the transition-metal-catalyzed activation of amides hinges upon the availability of operationally-simple, straightforward and high-yielding synthetic methods for the direct site-selective N-functionalization of the amide bond[16–19]. Broadly, this means that common primary and secondary amides, after simple N-functionalization, can be employed in a wide range of previously elusive transformations of high synthetic value, thus providing a major benefit to practitioners of synthesis.

We recently questioned whether more valuable transition-metal-free amidation reactions[24–28] could be employed in conjunction with the widely documented in the cross-coupling literature N-activation of the amide bond and the use of aromatic esters[16–21] to enable a broadly useful amide bond formation process via selective N–C and O–C cleavage. Herein, we report the successful realization of this concept and present the general method for the transition-metal-free transamidation of amides and amidation of esters by highly selective acyl cleavage with non-nucleophilic amines at room temperature. The protocol exhibits a remarkably broad scope across a range of structurally-diverse substrates (>75 examples) and is performed in an operationally-simple, environmentally-friendly manner, in particular in comparison to transition-metal-catalyzed and high-temperature methods[10,24–28]. We believe that this previously elusive type of transition-metal-free[29,30] functionalization of amides and esters by chemo- and regioselective N–C and O–C cleavage will be of great interest in the synthesis of pharmaceuticals, natural products, and polymers. It is worthwhile to note that all amides have been prepared directly from secondary and primary amides[22], thus representing an operationally-trivial and

**Previous work:**

**a** *Traditional transamidation*

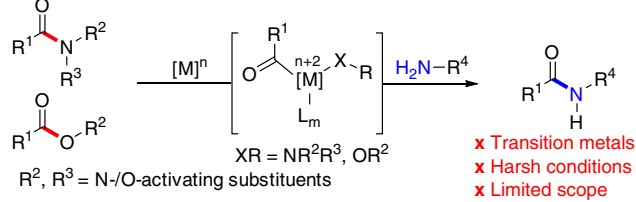

x Harsh conditions
x Challenging for 2° amides
x Challenging for non-nucleophilic amines

R$^2$, R$^3$ = aliphatic, Ar

**b** *Transition-metal-catalyzed cross-coupling*

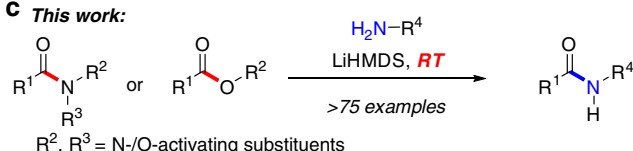

XR = NR$^2$R$^3$, OR$^2$

x Transition metals
x Harsh conditions
x Limited scope

R$^2$, R$^3$ = N-/O-activating substituents

**c** *This work:*

H$_2$N–R$^4$
LiHMDS, *RT*
>75 examples

R$^2$, R$^3$ = N-/O-activating substituents

● TM free ● mild conditions ● abundant and cheap reagents
● high selectivity ● operationally-simple ● broad scope

**Fig. 1** Recent directions in amide bond forming reactions. **a** Traditional transamidation. **b** Transition-metal-catalyzed N–C(O) and O–C(O) cross-coupling. **c** Transition-metal-free transamidation and amidation reactions by selective N–C/O–C cleavage

broadly applicable two-step method for transamidation of common primary and secondary amides under exceedingly mild conditions.

## Results

**Transamidation of amides.** In initial studies, we examined the transamidation of N-Boc activated secondary benzamide **1a**, which could be prepared in high yield by site-selective *N-tert*-butoxycarbonylation in high yield[22]. To our delight, we found that the proposed transition-metal-free transamidation proceeded in 94% yield on a gram scale in the presence of *p*-anisidine (2.0 equiv), while using LiHMDS (HMDS = hexamethyldisilazane, 3.0 equiv) as a base and toluene (0.25 M) as a solvent at ambient conditions. It is noteworthy that the reaction proceeded in high yield without the need for a toxic transition-metal-catalyst or strict precautions to exclude air (i.e., bench-top set-up), used inexpensive and readily available reagents[29,30], achieved the C–N bond cleavage with high selectivity, and advances the existing transamidation methods by utilizing common secondary amide as latent precursor. Further control experiments established that the base was required, with **2a** not formed in its absence. Furthermore, a survey of bases revealed that LiHMDS was optimal in terms of reactivity and C–N cleavage selectivity, while KO*t*-Bu, NaHMDS, *n*-BuLi and KHMDS provided inferior results (50–91%). The major side process of the transition-metal-catalyzed cross-coupling of amides[16–19], N-Boc scission, was not observed under the developed reaction conditions. Detailed optimization results are shown in Supplementary Table 1. We found that the reaction works efficiently with 1.5 equiv of amine and 1.5 equiv of LiHMDS in the model substrate. The reaction is sensitive to solvent concentration used. For some cases, the use of NaHMDS is preferred over LiHMDS. For reactive substrates, 1.0 equiv of amine suffices to achieve high conversion. The reaction was discovered during the development of broadly defined amidation protocols. A critical concern in transition-metal-catalysis

involves toxic and expensive reagents as well as the removal of undesired impurities from the post-reaction mixture. We recognized that a transition-metal-free protocol would provide a simple and powerful amidation method.

Examination of the scope revealed that a remarkably broad range of non-nucleophilic amines and amides are suitable for this mild, transition-metal-fee transamidation protocol (Fig. 2). Electronically-diverse and extremely sterically-hindered anilines were found to be excellent substrates. Notably, the reaction tolerates a wide range of challenging and sensitive substrates that

are incompatible or problematic with metal-catalyzed and high-temperature protocols, including esters, halides, heterocycles, highly electron-deficient arenes (e.g., the classically-challenging in cross-coupling pentafluoroaniline and 8-aminoquinoline)[31] as well as N,N-disubstituted anilines and deactivated aliphatic amines. Perhaps equally remarkably, a broad spectrum of amides, including a comprehensive survey of aliphatic amides with linear, branched and steric-hindrance at the α-carbon, can be employed in this protocol even though transition-metal-catalyzed reactions often require extensive ligand and reaction condition

**Fig. 2** Transition-metal-free transamidation of amides: reaction scope. Amide (1.0 equiv), **2** (2.0 equiv), LiHMDS (3.0 equiv), toluene (0.25 M), 23 °C, 15 h. Isolated yields. [a]**2** (1.5 equiv), LiHMDS (2.3 equiv). [b]DMF (0.25 M). [c]NaHMDS (3.0 equiv)

optimization[25], thus representing a significant advantage of the present method.

The fact that valuable amides are formed directly under mild, transition-metal-free conditions in this protocol is readily appreciated given that previous similar examples required expensive, toxic and sensitive Ni(0) and Pd(II)-catalyst systems[24–28], which additionally have the potential issues that the removal of undesired metal contamination would create a considerable problem for industrial research[29,30].

**Amidation of esters.** We were pleased to learn that the developed amidation method could be extended to aromatic esters (Fig. 3). Seminal studies by Yamaguchi and Itami[20], and more recently Newman[21] established the synthetic utility of O–C(acyl) cleavage in aryl esters by transition metals. The engagement of phenyl esters in transition-metal-catalyzed protocols allows the bench-stable, orthogonal ester moieties to be used directly as leaving groups. The process described here (Fig. 3) provides for a transition-metal-free amidation approach to amide bond formation from phenyl esters. Notably, the scope of the reaction is very broad and accommodates

**Fig. 3** Transition-metal-free amidation of esters: reaction scope. Ester (1.0 equiv), **2** (2.0 equiv), LiHMDS (3.0 equiv), toluene (0.25 M), 23 °C, 15 h. Isolated yields. [a]**2** (1.0 equiv), LiHMDS (2.0 equiv). [b]NaHMDS (3.0 equiv)

significant structural and electronic diversity in terms of both amine and ester components. Of particular note, the formation of sterically-hindered, electrophilic, highly electron-deficient, and aliphatic amides was achieved in high yields, thus further demonstrating the mild conditions and advantages of the present protocol. Collectively, the results in Figs. 2, 3 show that (1) the concept of a unified reactivity scale for manipulation of amides and esters by resonance destabilization is not limited to transition-metal-catalysis[16,32], (2) useful levels of selectivity superseding transition-metal-catalysis on several levels[16–21,29,30] can be achieved in the transition-metal-free amidation reactions of amides and esters by N–C and O–C cleavage.

**Transamidation of N,N-Boc$_2$ amides.** With respect to generality, efficient transamidation with non-nucleophilic amines was also observed in the case of N,N-Boc$_2$ amides (Fig. 4)[22]. In this case, the process significantly expands the known methods for trans-amidation of the primary carboxamide group to operationally-convenient, exceedingly mild reaction conditions, high functional group compatibility, and the use of pharmaceutically-valuable deactivated anilines. Several additional points with respect to the

substrate scope should be noted: (1) while this study is focused on the use of non-nucleophilic aromatic amides due to the importance of aryl amides in chemistry, we found that diethylamine is a competent nucleophile in amidation of esters (**4a**, 65%, unoptimized yield). (2) Moreover, we demonstrated that n-butylamine is a competent nucleophile in transamidation of amides under these conditions (**1a**, 97% unoptimized yield). (3) Intriguingly, the present method can be used to cleave N-8-aminoquinolinyl amides (88%, unoptimized yield) and the activated Yu-Wasa auxiliary (82%, unoptimized yield)[33].

**Transamidation of various amides.** Notably, the transamidation reaction is tolerant of various amide substitution patterns, including secondary N-alkyl amides, N-sulfonamide activation, and N-acyl-pyrroles, attesting to the generality of the protocol (Fig. 5)[16]. The functional group tolerance to aryl bromide should be noted. Since N-acylpyrroles can be readily prepared from primary benzamides[34], the latter process provides a valuable alternative two-step approach to transamidation of primary amides.

**Fig. 4** Transition-metal-free transamidation of N,N-Boc$_2$ amides: reaction scope. Amide (1.0 equiv), **2** (2.0 equiv), LiHMDS (3.0 equiv), toluene (0.25 M), 23 °C, 15 h. Isolated yields. [a]NaHMDS (3.0 equiv)

Pleasingly, chiral amino acids can be directly employed without detectable loss of enantiopurity (Fig. 6). It is particularly notable that the previous state-of-the-art example involved a transition-metal-catalyzed coupling of **4** with PhNH₂ at 110 °C[26], clearly establishing a significant value of the present method from the practical and environmental standpoint[29,30].

One-pot amide N-activation/amidation is also feasible (Fig. 7). This operationally-simple procedure was carried out using electronically deactivated, halide-containing aniline that (1) features electrophilic handles for classical cross-coupling, and (2) would not be tolerated in related metal-catalyzed amidation protocols[10,24–28].

The suitability of the transition-metal-free amidation in a model system for polymer synthesis was investigated (Fig. 8). Recently, Hillmyer and co-workers accomplished radical polymerization/post-polymerization metal-free transamidation of activated poly(N,N-Boc₂-amide)[15]. We established that the transamidation of a model N,N-Boc₂-isobutyramide bearing α-branching at the carbon center with aniline proceeded in high yield, thus demonstrating the enticing potential for modification in polymer synthesis.

The practical value of the method was further showcased in the synthesis of pharmaceuticals, a reversible MAO inhibitor, moclobemide and lidocaine, a high profile local anesthetic (Fig. 9)[35]. A defining feature in the approach to lidocaine is the potential of this protocol to provide an alternative disconnection owing to the stability of the ester group to streamline the synthesis of analogs. As shown in Fig. 9, the classic approach to lidocaine anesthetics involves a direct acylation of 2,6-dimethylaniline with 2-chloracetyl chloride, followed by an S$_N$2 reaction with diethylamine. The use of activated esters enables reversal of this sequence facilitating rapid synthesis of amide analogs of the powerful anesthetic.

**Mechanistic investigation**. Preliminary mechanistic studies were conducted. The results of mechanistic studies are summarized in Supplementary Figs. 1–4. The studies demonstrate that the reaction pathway involves initial deprotonation, followed by nucleophilic addition to the acyl bond. This mechanism is distinct from the traditional nucleophilic addition to tetrahedral intermediates, which proceed through addition-deprotonation pathway. As a key consideration, high selectivity of the acyl N–C/O–C cleavage under exceedingly mild conditions is achieved through ground-state destabilization of amide and ester bonds.

Specifically, intermolecular competition experiments revealed that the electronic nature of aniline does not significantly affect the reactivity; however, a trend favoring electron-deficient anilines should be noted (4-CF₃ vs. 4-MeO, amide: 3.6:1; ester: 1.3:1), while electron-deficient amides and esters are more reactive than their electron-rich counterparts (4-CF₃ vs. 4-MeO, amide: 2.3:1; ester: 5:1). Further competition experiments established a similar order of reactivity of amides and esters (N,N-Ph,Boc ≈ OPh, 1:1). Finally, anilines were found to be significantly more reactive than aliphatic amines (PhNH₂ vs. n-BuNH₂, amide: 9:1; ester > 20:1). Overall, these preliminary studies are consistent with the key role for amine deprotonation and, importantly, identify a common manifold for transition-metal-free functionalization of amides and esters. We hypothesize that aniline activation by hydrogen bonding to the carbonyl substrate activates the amine towards smooth deprotonation. The high selectivity of the nucleophilic addition hinges upon resonance destabilization of the acyl group[22,23]. Further studies to elucidate the mechanism are ongoing.

**Discussion**

In summary, we have developed a mild, transition-metal-free and operationally-simple protocol for transamidation of amides and

**Fig. 5** Transition-metal-free transamidation of amides. Amide (1.0 equiv), **2** (2.0 equiv), LiHMDS (3.0 equiv), toluene (0.25 M), 23 °C, 15 h. Isolated yields. [a]**2**: p-Anisidine (2.0 equiv)

**Fig. 6** Enantiopure amino-acid derivative. Reaction of enantiopure amino-acid derivative **4j** with aniline **2o**

**Fig. 7** One-pot N-activation/transamidation. Reaction of secondary amide **1r** with Boc$_2$O and aniline **2p**

**Fig. 8** Post-polymer modification. **a** Previous study using nucleophilic amines. **b** This study using non-nucleophilic amines in a model system

**Fig. 9** Synthesis of Moclobemide and Lidocaine. Reaction of ester **4k** with amine **2q**, and reaction of ester **4l** with aniline **2f**

amidation of esters. The scope of this transformation is very broad, the reagents that are utilized are cheap and readily available, and there is no need for toxic transition-metals or harsh reaction conditions. The practical value for applications in organic synthesis has been demonstrated through high functional group compatibility, preparation of pharmaceutical agents, potential for late-stage modification, and compatibility with enantiopure amino acids.

More generally, the concept deployed here advances the repertoire of transamidation methods to exceptionally mild and powerful amide bond forming reactions exploiting highly chemoselective, transition-metal-free N–C cleavage. Equally important is the pivotal role of resonance-guided common amide and ester manifold in transition-metal-free transformations in organic synthesis. Given

the importance of amide bonds in modern chemistry, we anticipate that this general and broadly applicable concept will be of great interest in the synthesis of functionalized amide-containing building blocks. Studies to expand the scope of the protocol with respect to the O-leaving group are underway.

## Methods

**General procedure for transamidation and amidation reactions**. An oven-dried vial equipped with a stir bar was charged with an amide or ester substrate (neat, 1.0 equiv), amine (typically, 2.0 equiv) placed under a positive pressure of argon, and subjected to three evacuation/backfilling cycles. Toluene (typically, 0.25 M) and LiHMDS (1.0 M in THF, typically, 3.0 equiv) were sequentially added with vigorous stirring at room temperature, and the reaction mixture was stirred at room temperature for an indicated time. After the indicated time, the reaction mixture was quenched with NH$_4$Cl (aq., 1.0 M, 1 mL), diluted with CH$_2$Cl$_2$ (10 mL), the organic

layer was washed with water (1 × 10 mL), brine (1 × 10 mL), dried and concentrated. Purification by chromatography on silica gel (EtOAc/hexanes) afforded the title product.

## Data availability

Experimental procedures and characterization data are available within this article and its Supplementary Information. Data are also available from the corresponding author on request.

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

## Acknowledgements

The NSF (CAREER CHE-1650766) is gratefully acknowledged for support. We thank Mr. Guang Hu and Prof. Stacey Brenner-Moyer (Rutgers University) for assistance with HPLC measurements.

## Author contributions

M.S. and G.L. conceived the concept and designed the experiments. G.L. conducted the chemical reactions described in the manuscript. M.S. wrote the manuscript and all authors contributed to the reading and editing of the manuscript. G.L. compiled the Supplementary Information.

## Additional information

**Competing interests:** The authors declare no competing interests.

