## [Peer Review File · Nature Communications]

Reviewer #1 (Remarks to the Author):

The group of Szostak has in recent years contributed significantly to the research regarding amide activation, by development of a wide range of synthetic protocols that enables various transformations of amides. In this paper, Li and Szostak follows up their excellent work with a modified and improved protocol for the transamidation/amidation of a variety of amides/esters, which not only displays a remarkable substrate scope but also proceeds efficiently at room temperature. Although, this work fundamentally builds on concepts previously described by the group of Szostak, I still believe that it is suitable for publication in Nature Communications, given the usefulness of this methodology and the immense interest in amide transformation protocols.

Although, I recommend publication of this work, there are a couple of points that should be addressed before it is published.

- 1) Figure 1. To avoid a situation where apples are compared to oranges, R2 and R3 should be specified, as some protocols transform amides with mainly aliphatic/aromatic N-substituents, while some other methods rely on pre-functionalized starting materials with sterically-hindered or other activating N-substituents that disturb the amide resonance and activate the C-N bond.
- 2) Regarding the amidation of esters, I do wonder how tolerant the present protocol is towards different oxygen-substituents. All examples in Figure 3 make use of $-OPh$, which is perhaps not ideal given that this requires handling and the liberation of phenol—a very toxic chemical. Would this protocol also work for $-OBn$ or $-OEt$, which releases more benign byproducts? Showing that these ester substrates are within the scope of the method would also help to further demonstrate the usefulness of this protocol. If these substrates are beyond the scope of the present protocol, it should be highlighted in the main text.
- 3) For the transamidation or the ester amidation reactions, is it possible to use ammonia or dimethylamine (or diethylamine) as the amine nucleophile?
- 4) In Figure 5, the authors demonstrate that the method is also applicable to substrates with a variety of N-substituents. In a recent publication, which could perhaps be worth citing, Verho and co-workers (*J. Org. Chem.*, 2018, 83 (8), pp 4464–4476) demonstrate a metal and additive-free protocol for the mild cleavage of the 8-aminoquinoline directing group from a number of complex substrates. Would the present method also be applicable for cleavage of directing groups, such as 8-aminoquinoline and the Yu-Wasa auxiliary?

5) Finally, in Figure 9c, I am not sure the advantages of the streamlined approach vs the classic disconnection is clearly highlighted. Comparing the two, the classic disconnection would involve a direct acylation of 2,6-dimethylaniline with 2-chloroacetyl chloride, which is a reaction that has been reported to be quantitative. Thereafter the final -Cl is replaced in an Sn2 reaction with diethylamine, which is also known in the literature to be quantitative. In comparison, the streamlined approach would first involve the synthesis of the highlighted phenyl ester from two separate reactions (analogous acylation and Sn2 reaction reactions). Then a third step would be needed to amidate the highlighted ester (so it seems longer...)! Here, the authors should first of all present the entire synthetic sequence to lidocaine, and show the yields for all three steps. Then, it could perhaps be good to further elaborate the discussion regarding Figure 9, comparing the classic and streamlined approach in greater detail. Motivating advantages/disadvantages of the two methods.

Reviewer #2 (Remarks to the Author):

Authors describe an original and powerful methodology for trans-amidation of amides and amidation of esters. The manuscript is well written and clearly presented. Authors provide a general and broadly applicable synthesis of molecules containing amide bond.

Despite its novelty, several essential points have to be discussed in main text.

It is important for reader to understand how the reaction was discovered and optimized. There is a lack of information concerning the study of the reaction development (for transamidation of amides and esters)

One negative point of this methodology is that it used 2 equivalent of amine partner (the unreacted amine could be reused ? degraded ?) and 3 equivalent of LiHMDS. Is all this excess is consumed ? Degraded ? Without solvent, do this reaction could work neat ?

For synthesis of pharmaceuticals in figure 9, reaction conditions have changed without any explanation (2 quiv of NaHMDS instead of 3 equiv of LiHMDS, and 1 equiv of amine instead of 2 equiv of amine). Authors should mention it on main text and they should explained theses changes. Conditions in figure 9 constitute much better suitable conditions than previous one used in this manuscript.

Theses points have to be discussed in main text, not in SI.

Moreover, authors claimed mechanistic studies in SI, but theses experiences do not allow any significant advances in the mechanism understanding. This is not totally convincing.

In SI, no HRMS nor melting points measurement were reported. There are some spectra that have some non-attributed pics (as for 3b, 3d, 3k, 3z, 3ag,): traces of solvent (DMSO, H₂O...) ? Impurities ? It has to be mentioned directly on spectra. In some cases it is not negligible and it surely gave wrong yields. 3am is not pure.

For all points mentioned above, this manuscript could not be accepted for publication in Nature Communication.

Reviewer #3 (Remarks to the Author):

Szostak and co-workers describe a metal-free transamidation of amides and an amidation of esters using same reactions conditions. They consist of the use of a base which mediates the amidation process. The simplicity of the procedure and the wide functional group tolerance of the reaction conditions are really interesting. Indeed, the authors highlight very well the usefulness of this methodology by applying this method on more than 70 examples in point of view of functional groups tolerance but also in point of view of synthetic methodology. Remarkably, this procedure is compatible with N-Boc-benzamides as well as N-Boc-aliphatic amides derivatives. Very challenging substrates such as sterically hindered α,α -branched N-Boc amides are converted with remarkably high yields. The method is also compatible with non-nucleophilic amines such as anilines type substrates as much as aliphatic amines. The authors show also that the reactions conditions are able to convert phenyl ester in amides with the same functional groups tolerance and high selectivity as well as N,N-Boc²-amides, sulfonyl activated amides and N-acyl pyrroles. All of this shows the high generality of this original process to synthesize amides.

Appreciated efforts are done to display the synthetic usefulness of this methodology in synthesis of enantiopure compounds (no epimerization is observed), post-polymer modification (model substrates are functionalized) and medicinal drugs synthesis (2 active compounds are synthesized).

Finally, some interesting preliminary mechanistic studies are done.

Therefore, I strongly recommend this manuscript for publication in nature communications.

Point-by-point response to the referees' comments

As requested by the Reviewer no. 1:

The Reviewer no. 1 was very positive about the manuscript. We thank the reviewer for all comments and suggestions. As suggested by this reviewer, we have addressed the following points in the manuscript:

1. Figure 1 has been revised as suggested. R2 and R3 have been specified.
2. The reviewer suggested using –OBn and –OEt oxygen-substituents in the protocol. We agree with the reviewer that the use of aliphatic esters in amidation reactions would be another welcome addition to the synthetic toolbox. While our studies focused on the use of –OPh leaving group because of the utility of this leaving group in organic chemistry and the well-established potential of phenolic esters in cross-coupling, our preliminary results indicate that amidation of aliphatic esters might be possible. We are currently evaluating this process and the results will be disclosed when the data is available. This point has been clarified in the manuscript.
3. We have tested the use of diethylamine as the amine nucleophile as suggested. While our study was centered on the use of aromatic amines due to the importance of aryl amides in chemistry and general lack of methods for their synthesis from amides and esters, our results indicate that diethylamine is a competent nucleophile in amidation of esters (65% unoptimized yield). Moreover, we demonstrated that n-butylamine is a competent nucleophile in transamidation of amides (97% unoptimized yield). These points have been clarified in the manuscript.
4. The suggested publication has been cited. The reviewer suggested testing cleavage of directing groups, such as 8-aminoquinoline and the Yu-Wasa auxiliary. We thank the reviewer for this suggestion. In our preliminary studies we demonstrated that indeed the method could be used to cleave N-8-aminoquinolinyl amides (88% unoptimized yield) and the activated Yu-Wasa auxiliary (5-F-C₆H₄-amides, 82% unoptimized yield). These points have been clarified in the manuscript.
5. Figure 9c has been revised as suggested by the reviewer. The entire synthetic sequence to lidocaine has been presented, discussion including motivating advantages/disadvantages has been added.

As requested by the Reviewer no. 2:

The Reviewer no. 2 was very positive about the manuscript. We thank the reviewer for all comments and suggestions. As suggested by this reviewer, we have addressed the following points in the manuscript:

1. A discussion on the reaction discovery and optimization has been added.
2. Details on the reaction optimization have been added. The reaction works efficiently with 1.5 equiv of the amine and 1.5 equiv of LiHMDS in the model substrate. The reaction is sensitive to solvent concentration used. For some cases, the use of NaHMDS is preferred over LiHMDS. For reactive substrates, 1.0 equiv of amine suffices to achieve high conversion. These points have been clarified in the manuscript.
3. The mechanism of the method has been addressed and the pathway clarified.
4. All substrates have been characterized according to the established standards. This information has been clarified in the SI.

5. Traces of water and residual solvent from DMSO-d₆ in 3b, 3d, 3k, 3z, 3ag have been clarified, 3am exists as a mixture of rotamers. These points have been clarified.

As requested by the Reviewer no. 3:

The Reviewer no. 3 was very positive about the manuscript. We thank the reviewer for all comments and suggestions.

Reviewer #1 (Remarks to the Author):

I am very satisfied to see that the revised manuscript goes to great extent when addressing the comments I raised in my first review. The extensions of the scope and the clarifications of certain figures have now settled my concerns, and I now wholeheartedly recommend this work for publication immediately, without any further changes.

Reviewer #2 (Remarks to the Author):

All recommendations were followed

This communication can be published in nature communications in its actual form.

Point-by-point response to the referees' comments

We thank the reviewers for all comments and suggestions during the submission process.